# Overexpression *Bombyx mori* HEXIM1 Facilitates Immune Escape of *Bombyx mori* Nucleopolyhedrovirus by Suppressing BmRelish-Driven Immune Responses

**DOI:** 10.3390/v14122636

**Published:** 2022-11-25

**Authors:** Guanping Chen, Yuedong Li, Xiangshuo Kong, Shudi Zhao, Jiale Li, Xiaofeng Wu

**Affiliations:** 1College of Animal Sciences, Zhejiang University, Hangzhou 310058, China; 2Key Laboratory of Silkworm and Bee Resource Utilization and Innovation of Zhejiang Province, Hangzhou 310000, China

**Keywords:** BmNPV, BmHEXIM1, immunity escape, viral proliferation

## Abstract

*Bombyx mori* nucleopolyhedrovirus (BmNPV), a typical arthropod-specific enveloped DNA virus, is one of the most serious pathogens in silkworm farming, but the potential mechanisms of the evasion of innate immune responses from BmNPV infection are still poorly understood. HEXIM1 is an RNA-binding protein, best known as an inhibitor of positive transcription elongation factor b (P-TEFb), which controls transcription elongation by RNA polymerase II. In this study, *Bombyx mori* HEXIM1 (BmHEXIM1) was cloned and characterized, and its expression was found to be remarkably upregulated after BmNPV infection. Furthermore, BmHEXIM1 was detected to increase the proliferation of BmNPV, and its full length is essential for assisting BmNPV immune escape by suppressing BmRelish-driven immune responses. This study brought new insights into the mechanisms of immune escape of BmNPV and provided theoretical guidance for the breeding of BmNPV-resistant silkworm varieties.

## 1. Introduction

Silkworm (*Bombyx mori*) is a model organism of lepidopteran insects and one of the economic insects that have contributed greatly to the sericultural industry. It is also an important genetic research model after *Drosophila* and has a good scientific research background for exploring the mechanism of interaction between insects and their pathogens. *Bombyx mori* nucleopolyhedrovirus (BmNPV), a virulent virus, has brought great losses to the sericultural industry annually. After viral infection, biochemical and structural changes occur in the host cells, resulting in various physiological disorders. Therefore, probing the response of the silkworm to BmNPV infection and its mechanism of action is of great significance for not only the development of new prevention and control measures for BmNPV but also the improvement of effect of baculoviruses in controlling pest insects in agriculture and forestry.

The innate immune response is part of the overall immune system that protects the hosts from infection by pathogens [1]. Silkworms fight foreign microbial infections by initiating powerful immune responses that are mediated by the hemolymph, the fat body, the midgut, and other tissues. Foreign organisms that have entered the body of the insect are recognized by the immune system when pathogen-associated molecular patterns bind host pattern recognition receptors. This subsequently activates immune signaling pathways, amplifies the immune response, induces the production of cytokines, including antimicrobial peptides, and promotes inflammation to create a physical barrier that prevents the spread of infection. However, baculoviruses BmNPV have developed several strategies to evade the host’s immune system and promote their own replication and proliferation, including the inhibition of antiviral melanization, autophagy, apoptosis, RNAi, and regulation of the cell cycle. In the past decades, many cellular proteins that promote BmNPV proliferation have been identified, including *Bombyx mori* nuclear hormone receptor 96 (BmNHR96) [2], *Bombyx mori* receptor expression-enhancing protein (BmREEPa) [3], *Bombyx mori* E3 ubiquitin–protein ligase SINA-like 10 (BmSINAL10) [4], and the heat shock protein (HSP) family [5,6,7].

Hexamethylene bisacetamide-inducible protein 1 (HEXIM1) is an estrogen receptor disrupting protein that can be induced by hexamethylene bisacetamide (HMBA) and analogues [8], histone deacetylase inhibitors [9] and nucleotide depletion [10]. Moreover, HEXIM1 was initially shown to be associated with 7SK RNA, an abundant small nuclear non-coding RNA [11], with the primary function of sequestering CyclinT1/CDK9 in 7SK small nuclear ribonucleoprotein particles (snRNP) [12,13]. These interactions inhibit P-TEFb to prevent the transcription elongation by RNA polymerase II [14]. In addition to 7SK RNAs, HEXIM1 specifically binds and stabilizes a subset of mRNAs [10]. Recently, HEXIM1 was shown to bind the NEAT1 non-coding nuclear RNA to form a new RNA-protein complex [15]. Furthermore, HEXIM1 is also able to repress several other transcription factors, including NF-κB [16], glucocorticoid receptor (GR) [17], and estrogen receptor α [18]. However, the functions and mechanisms of the insect HEXIM1 involved in baculovirus life cycles remain unknown.

In this study, we found that *Bombyx mori* HEXIM1 (BmHEXIM1) expression was remarkably upregulated after BmNPV infection. In order to make clear the role of BmHEXIM1 in the viral life cycle, a series of experiments were conducted, including the knockdown and overexpression of BmHEXIM1, as well as subcellular localization assay. The results showed that BmHEXIM1 played an important role in the BmNPV life cycle, and further studies revealed that the overexpression of BmHEXIM1 suppressed host innate immunity, resulting in the promotion of BmNPV proliferation. These results provide valuable information for elucidating the molecular mechanism of BmNPV immune escape and lay a good scientific and theoretical foundation for the prevention and control of BmNPV infection.

## 2. Materials and Methods

### 2.1. Insect, Cell Culture, Virus, and Chemical Treatment of BmN Cells

The plasmids pIZ/V5-His and BmNPV were preserved in the laboratory. All recombinant viruses were previously constructed in our laboratory. BmN cells were preserved in the laboratory and maintained at 27 °C in Grace’s medium supplemented with 3% fetal bovine serum (FBS, Gibco, Waltham, MA, USA). BmN cells were cultured in the presence or absence of 1 µM phorbol 12-myristate acetate (PMA) (Beyotime, Shanghai, China) for 24 h as previous research performed [19]. In addition, HMBAs (Glpbio, Montclair, USA) were dissolved in culture medium at 200 mM [20,21]. Next, BmN cells were seeded at low confluence (about 5 × 10^3^ cm^−2^) and treated 12 h, 24 h, and 48 h later with HMBA. HMBA was used in the range 1–30 mM [22]. The silkworm larvae (commercial name Qingsong × Haoyue) were reared with fresh mulberry leaves at 25 °C temperature.

### 2.2. Phylogenetic Analysis

Homologue searches were performed in insects with BLASTP using the BmHEXIM1 homologue of silkworm as query sequence against databases of National Center for Biotechnology Information (NCBI, Available online: https://www.ncbi.nlm.nih.gov/, (accessed on 28 October 2022)) database. Sequence alignments were inferred using MUSCLE algorithm in MEGA6 based on a codon model [23]. A phylogenetic tree containing HEXIM1 sequences in Lepidoptera, Hymenoptera, and Diptera was built by neighbor joining (NJ) method in MEGA6, and relative support for the internal nodes in the trees was estimated by bootstrap method.

### 2.3. Tissue Sample Collection

On day one of the fifth instar, half of male silkworms were selected for the injection of the virus. After 72 h, tissue samples of all the silkworms were collected, including testis, middle silk gland, posterior silk gland, midgut, fat body, hemolymph, and Malpighian tube. Tissue samples were thoroughly washed in DEPC water and flash-frozen in liquid nitrogen. All the samples were kept at −80 ℃ until use.

### 2.4. The Construction of pIZ/V5-BmHEXIM1-Flag and Two Mutants Expression Vector

The BmHEXIM1 cDNA was amplified with primers (forward 5′ GAATTCATGGACGCGGATAATCTTATCATA 3′ and reverse 5′ TCTAGACTACTTGTCATCGTCGTCCTTGTAATCATCCTTAGTATGGAACCCATTCAC 3′) and cloned into the insect expression vector, PIZ/V5-His plasmid (Invitrogen, Carlsbad, CA, USA). The primers were flanked by restriction enzyme sites, EcoRI and XbaI (Takara, Dalian, China) and FLAG-tag. Eventually, we obtained the overexpression vector, pIZ/V5-BmHEXIM1-FLAG. Briefly, BmHEXIM1-1 and BmHEXIM1-2 were cloned from pIZ/V5-BmHEXIM1-FLAG and then cloned into the pIZ/V5-His plasmid.

### 2.5. RNA Extraction, cDNA Synthesis, and Quantitative Real-Time Polymerase Chain Reaction (RT-qPCR)

Total RNA was extracted from tissue samples and cells using Trizol reagent (Sangon Biotech, Shanghai, China) according to the manufacturer’s instructions. The qualities and concentrations of RNA samples were measured using NanoDrop 2000 spectrophotometer (ThermoFisher Scientific, MA, USA). Then, 1 μg of total RNA was used for the cDNA synthesis by use of PrimeScript RT reagent Kit with gDNA Eraser (TaKaRa, Dalian, China). All the cDNA solutions were diluted five-fold and stored at −20 °C for further use. The transcriptional level of genes was quantified by RT-qPCR with SYBR Premix ExTaq II (TaKaRa, Dalian, China) in a LightCycler 480 II real-time PCR system (BioRad, Shanghai, China). All gene expressions were normalized to the expression of BmRPL32 gene. The relative fold changes of gene expression were determined by the threshold cycle (2^-△△CT^) method [24]. All the primers used in this study are listed in Appendix A.

### 2.6. Immunofluorescence (IF) and Fluorescence Microscopy

BmN cells were plated onto 35 mm coverslips and transiently transfected using Lipo8000^TM^ Transfection Reagent (Beyotime, Shanghai, China) according to the manufacturer’s protocol. In all experiments, BmN cells were transfected with plasmids expressing FLAG- or HA-tags 1 d after plating. After 2 d post-transfection, cells were infected with BmNPV or not. Then, cells were fixed with 4% paraformaldehyde in PBS for 20 min, then permeabilized with 0.1% TritonX-100 for 15 min and blocked for 1 h with 1% bovine serum albumin (BSA). The cells were incubated with primary antibodies anti-FLAG (1:1000) or anti-HA (1:200) overnight at 4 ℃, then incubated with donkey anti-mouse IgG/TRITC (Sangon Biotech, Shanghai, China) and donkey anti-rabbit IgG/FITC (Sangon Biotech, Shanghai, China) according to the experiments. Next, cells were sealed with 4′,6-diamidino-2-phenylindole (DAPI, Beyotime, Shanghai, China) for 15 min and examined through ZEISS LSM 880 confocal scanning laser microscopy (CSLM).

### 2.7. RNA Interference Assay

The siRNAs that targeted the BmHEXIM1 gene were synthesized by Sangon Biotech (Shanghai, China), and a control siRNA was also provided by Sangon Biotech. siRNA transfections were performed using LipoRNAiTM Transfection Reagent (Beyotime, Shanghai, China) according to the manufacturer’s protocol. The sequences of siRNA were shown as follow:

siControl (sense: UUC UCC GAA CGU GUC ACG UTT; antisense: AAA CGT GAC ACG TTC GGA GAA).

siBmHEXIM1 (sense: GCG CGA AUA CCA AAC GUA ATT; antisense: AAT TAC GTT TGG TAT TCG CGC).

### 2.8. Co-Immunoprecipitation (Co-IP) Assay

To confirm the interaction between BmHEXIM1 and PK1 or viral polymerase subunits, HA-IP assay was carried out. BmN cells were transfected with pIZ/V5-BmHEXIM1-FLAG plasmids and infected with the recombinant virus at an MOI of 1. Infected cells were collected at 48 h and washed three times with PBS supplemented with protease inhibitor cocktail (Bimake, Beijing, China). The cell pellets were then resuspended in cell lysis buffer (Beyotime, Shanghai, China) supplemented with protease inhibitor cocktail. Next, cell lysates were incubated with mouse anti-HA immunomagnetic beads (Bimake, Beijing, China) at 4 °C overnight. After the incubation, beads were harvested with a magnetic separator and washed with PBS supplemented with protease inhibitor cocktail five times.

### 2.9. Western Blotting

After harvesting different group samples and quantifying the concentration by Bradford assay (BioRad, Shanghai, China), SDS-PAGE and transmembrane were as previously described [25].

### 2.10. Viral Titer Determination and Measurement of BmNPV Proliferation in BmN Cells

Viral growth curve analysis TCID_50_ end-point dilution was used to determine the production of infectious budded viruses (BVs) [26]. TCID_50_ was determined in triplicate by infecting BmN cells in 96-well plates and analyzed by fluorescent microscopy after incubation at 27 °C for 7 days. The green or red fluorescence intensity was employed to further evaluate the proliferation of BmNPV-eGFP or BmNPV-mCherry.

### 2.11. Statistical Analysis

Results from three independent experiments are presented as mean ± SD. Data were analyzed using a Student’s *t*-test for the comparison of two groups or a two-way ANOVA for comparison of multiple groups (GraphPad Prism 6.0, San Diego, CA, USA). The number of asterisks represents the degree of significance with respect to *p*-value. *P*-values were provided as * *p* < 0.05; ** *p* < 0.01; *** *p* < 0.001; **** *p* < 0.0001.

## 3. Results

### 3.1. Identification and Characterization of BmHEXIM1

To explore the host factors involved in the regulation of baculovirus transcriptional elongation, the expression of transcriptional elongation regulators was analyzed at different phases after BmNPV infection. As shown in Figure 1A, the expression of *Bombyx mori* SPT5 (BmSPT5) and *Bombyx mori* NELF-D (BmNELF-D), but not *Bombyx mori* CDK9 (BmCDK9), were upregulated after BmNPV infection. Noteworthily, the expression of BmHEXIM1 significantly increased (Figure 1A).

The coding sequence of BmHEXIM1 is 996 bp and encodes 332 amino acids. Moreover, a query of the NCBI database showed that BmHEXIM1 was located on chromosome 25 of the silkworm. By BLASTP analysis, the E-value and identity HEXIM1 sequence of *Bombyx mori* (accession XP_004921854.1) were found to be significantly different compared with HEXIM1 from *Drosophila melanogaster* (accession NP_650381.1 and NP_731932.1), *Mus musculus* (accession NP_620092.1), and *Homo sapiens* (accession NP_006451.1), indicating it may be a distinct lineage (Appendix A). Furthermore, the conserved motifs in the HEXIM1 of these species were analyzed, and a total of four motifs were identified (Appendix A). In addition, the HEXIM1 phylogenetic tree showed that lepidopteran HEXIM1 were linked together with a high confidence level, whereas *Hymenopteran* and *Dipteran* HEXIM1 were distantly related to them as the outgroup (Appendix A).

### 3.2. BmHEXIM1 Showed a Significant Response to BmNPV Infection

The expression of BmHEXIM1 in different tissues of silkworm larvae was analyzed, and the results illustrated that the expression of BmHEXIM1 greatly increased after viral infection in most of larval tissues, including the testis, middle silk gland, and fat body (Figure 1B). In addition, the subcellular localization of BmHEXIM1 showed that, like mammals, BmHEXIM1 was also mainly accumulated in the nucleus and less distributed in the cytoplasm (Figure 1B). Interestingly, BmHEXIM1 was translocated from the nucleus to the cytoplasm at 24 h of viral infection, and then transferred to the nucleus and, finally, accumulated in the virogenic stroma (VS) region (Figure 1C), in which viral DNA replication and nucleocapsid assembly occur [27,28,29]. These results implied that BmHEXIM1 may play a potential role in viral infections, especially in the late phase.

### 3.3. Knockdown of BmHEXIM1 Significantly Suppressed BmNPV Proliferation

To confirm whether BmHEXIM1 is involved in BmNPV infection, the exogenous siRNA-mediated knockdown of BmHEXIM1 was performed, and it was confirmed that siRNA-BmHEXIM1 significantly reduced its expression (Figure 2A). After the knockdown of BmHEXIM1, BV titers were reduced by about 95% at 24 h and by about 50% at 48 h (Figure 2B). Furthermore, fluorescence microscopy analysis showed that eGFP-positive cells in the experimental group were less than those in the negative control (Figure 2C), indicating that BmHEXIM1 is important for the proliferation of BmNPV. Subsequently, the expression of viral genes was analyzed, and the results showed that the expression of viral genes were greatly down-regulated (Figure 2D). Moreover, BmNPV F-like protein Bm14, a functional envelope fusion protein of BVs [30], was found to be suppressed after the knockdown of BmHEXIM1 (Figure 2E). The above results indicated that the knockdown of BmHEXIM1 significantly suppressed BmNPV proliferation.

### 3.4. Overexpression of BmHEXIM1 Significantly Promoted BmNPV Proliferation

As shown in Figure 3A, the Western blot result showed that the BmHEXIM1 was continuously expressed, indicating that the overexpression vector pIZ/V5-BmHEXIM1 works well, as expected. The overexpression of BmHEXIM1 resulted in a 2.3-fold increase in BV titers at 24 h and a 9.8-fold increase at 48 h (Figure 3B). Moreover, fluorescence microscope analysis indicated that the number of mCherry-positive cells in the experimental group was more than that of the negative control (Figure 3C). In contrast to the results of the knockdown of BmHEXIM1, the overexpression of BmHEXIM1 promoted the expression of these viral genes (Figure 3E), and Western blot assay showed that the expression of Bm14 was severely upregulated (Figure 3F).

### 3.5. The Full Length of BmHEXIM1 Was Essential for Promoting BmNPV Proliferation

It was demonstrated that HEXIM1 is a complex protein possessing several regions to exert multiple functions [31], and BmHEXIM1 contains the HEXIM functional domain (P53-L177). To identify the functional domains of BmHEXIM1 responsible for promoting BmNPV proliferation, two mutants with or without HEXIM functional domain were constructed, named BmHEXIM1-1 and BmHEXIM1-2, respectively (Figure 3D). However, neither the overexpression of BmHEXIM1-1 nor BmHEXIM1-2 could achieve as many effects in promoting viral proliferation as the full length of BmHEXIM1, but BmHEXIM1-1 appeared to be more important in promoting BmNPV proliferation (Figure 3E), and this result was further confirmed by the detection of Bm14 by Western blot (Figure 3F). Taken together, the above experiments showed that the full length of BmHEXIM1 is important for its function in promoting viral proliferation.

### 3.6. BmHEXIM1 Is Not Involved in Controlling Viral Transcriptional Elongation by Regulating Viral Polymerase Phosphorylation

During transcriptional elongation, the phosphorylation of HEXIM1 by cAMP-PKA signaling results in the release of its inactive P-TEFb/HEXIM1/7SK snRNP complex for transcriptional elongation [32]. In addition, PK1, a protein kinase, is involved in the regulation of viral gene expression as a transcription factor in the very late phase [33] and it is also involved in the phosphorylation of viral RNA polymerase [34]. We, thus, hypothesized the potential role of BmHEXIM1 in regulating viral RNA polymerase phosphorylation. The co-expression of BmHEXIM1 and PK1, or two subunits of viral RNA polymerase, P47 and LEF-9, led to dramatic changes in the intracellular distribution of BmHEXIM1 and colocalized to the cytoplasm and cytoplasmic membrane, although BmHEXIM1 could be detected in the vs. region (Appendix A). However, we did not observe the direct interaction of BmHEXIM1 with PK1 and viral RNA polymerase by co-IP (Appendix A). These results suggest that BmHEXIM1 promoted viral proliferation not by regulating viral polymerase phosphorylation that regulates viral transcriptional elongation but by other pathways.

### 3.7. Overexpression of BmHEXIM1 Contributed to BmNPV Immune Escape by Suppressing BmRelish-Driven Immune Responses

HEXIM1 is a transcriptional regulator that functions as a general RNA polymerase II transcriptional repressor and also regulates NF-κB-, ESR1-, NR3C1-, and CIITA-dependent transcriptional activity [13,16,35]. In addition, the silkworm’s immune system is activated upon BmNPV infection, and *Bombyx mori* Relish (BmRelish), an orthologue of NF-κB, and *Bombyx mori* STING (BmSTING) are indispensable for silkworm antiviral responses [36,37]. Moreover, *Bombyx mori* cecropin A (BmcecA) and *Bombyx mori* cecropin B (BmcecB), two antimicrobial peptide genes, are the downstream factors of the BmRelish-mediated pathway. Interestingly, the overexpression of BmHEXIM1 suppressed the expression of BmSTING and BmRelish (approximately 70–80%) (Figure 4A,B). Meanwhile, both BmcecA and BmcecB were also down-regulated about 50% (Figure 4C,D). Although both mutants repressed the expression of these genes, the effect was inferior to that of the full length of BmHEXIM1 (Figure 4A–D), suggesting that the full length of BmHEXIM1 is essential for exerting its function. However, the expression of these four important genes was not affected by the knockdown of BmHEXIM1 (Figure 4A–D), implying that a low-level expression of BmHEXIM1 did not affect the induction of host immune pathways. In addition, the phosphorylation level of RNP Ⅱ was detected by western blot, and the results indicated that the phosphorylation level of RNP Ⅱ was not affected by BmHEXIM1 and mutant proteins (Figure 4E). Briefly, the overexpression of BmHEXIM1 may specifically regulate BmSTING, BmRelish, and BmRelish target gene expression, thereby suppressing the immune response during BmNPV infection.

There was evidence that HEXIM1 plays an inhibitory role in NF-κB-dependent gene transcription [16]. To further assess whether such an effect of BmHEXIM1 takes place at the transcription level, mRNA expression was assayed after stimulation with PMA, a specific activator of Protein kinase C (PKC) and, hence, activates NF-κB [19]. Stimulation with PMA resulted in a three-fold increase in BmRelish expression and about a four-fold increase in BmRelish target gene expression (Figure 4F–I). However, the effect of PMA stimulation was inhibited in the presence of BmHEXIM1 (Figure 4F–I), indicating that BmHEXIM1 acted at a transcriptional level and repressed BmRelish-dependent transcription. Briefly, BmHEXIM1 assisted BmNPV immune escape by suppressing BmRelish-dependent transcription but required intact BmHEXIM1.

### 3.8. HMBA Induced BmHEXIM1 Expression Leading to the Promotion of Viral Proliferation

HMBA is a small molecule that has been investigated by the National Cancer Institute, and the overexpression of HEXIM1 was accomplished by treatment with HMBA, which induces HEXIM1 expression [38,39]. Therefore, we predicted that HMBA would induce BmHEXIM1 expression. As we expected, BmHEXIM1 expression was upregulated by HMBA stimulation, and the expression of BmHEXIM1 was upregulated two-fold under 10 mM HMBA-inducing conditions (Figure 5A). Further studies showed that, although the expression of BmHEXIM1 increased after different temporal 10 mM HMBA treatment, BmHEXIM1 expression was upregulated two-fold after 24 h of treatment, which is more suitable for the following experiments (Figure 5B). In addition, the expression of viral genes was significantly upregulated after the cells were treated with HMBA (Figure 5C). Moreover, the expression of Bm14 in HMBA-treated cells was also significantly increased (Figure 5D). Next, the effect of HMBA on BmRelish-dependent transcription and the expression of BmRelish target genes were tested. BmSTING and BmRelish were down-regulated by almost 50% as a result of HMBA (10 mM) treatment for 24 h (Figure 5E). A down-regulation of approximately 55% for BmcecA and 50% for BmcecB expression was confirmed by quantitative real-time PCR (Figure 5E). Briefly, HEXIM1, which is induced after treatment with HMBA, acts as a suppressor of BmRelish, leading to the promotion of viral proliferation.

## 4. Discussion

As a well-studied arthropod-specific double-stranded DNA virus, BmNPV relies on host cell machinery to meet its needs for the viral replication, production, and dissemination of progeny virions. Our results revealed that BmNPV infection triggered the upregulation of BmHEXIM1 and altered its subcellular localization. A previous study showed that HEXIM1 ubiquitination sequestered it in the cytoplasm [40] and that the ubiquitin–proteasome system was required for BmNPV to establish infection in the early phase [41], but whether early viral proteins with ubiquitin ligase activity, such as IE2 and ubiquitin, alter subcellular localization of BmHEXIM1 requires further characterization. In addition, the knockdown of BmHEXIM1 resulted in suppressing the expression of viral genes, leading to low levels of viral proliferation. In contrast, when BmHEXIM1 was stably overexpressed, BmNPV proliferation was increased. Noteworthily, HEXIM1 is best known as the inhibitor of positive transcription elongation factor b (P-TEFb), which controls the transcription elongation of RNA polymerase II and the Tat transactivation of human immunodeficiency virus [31]. However, BmHEXIM1 does not directly interact with viral RNA polymerase, implying that BmHEXIM1 does not promote viral proliferation by regulating viral transcriptional elongation. Further studies showed that BmHEXIM1 was involved in assisting BmNPV immune escape.

The host-BmNPV interaction is essentially like a tug of war, with the virus eager to propagate and spread and the host striving to eliminate foreign material to maintain equilibrium during infection [42]. Therefore, upon viral invasion, the host activates the innate immune response and rapidly establishes antiviral defense mechanisms. The IMD and Toll signaling pathways participate in insect antiviral immune responses [43,44], and the expression of the IMD pathway-associated protein BmRelish can directly reflect the level of the IMD pathway [42]. Moreover, BmNPV infection induces cyclic GMP-AMP (cGAMP) production in BmE cells, and BmSTING responds to cGAMP and activates the Dredd-caspase-mediated NF-κB (BmRelish) antiviral signaling pathway [37]. Furthermore, Relish also positively regulates the expression of STING [45]. Despite BmNPV infection activating the host immune-system-induced antiviral response, the virus persists in the host and escapes immune control as evidenced by the generation of new viral particles after BmNPV infection in immunocompetent silkworms. Immune escape contributes to viral persistence, yet little is known about baculovirus BmNPV. Mechanistically, our preliminary findings suggest that overexpression of BmHEXIM1 could suppress BmRelish-driven immune responses during BmNPV infection. However, the knockdown of BmHEXIM1 did not affect BmRelish-driven immune responses and may inhibit BmNPV proliferation by other molecular mechanisms, which needs further investigation. It is worth mentioning that the involvement of antimicrobial peptides in the regulation of an antiviral response has also recently been reported [46,47], especially BmcecA and BmcecB, which are regulated by BmRelish, played an important role in anti-BmNPV [48]. In addition, HMBA was an inducer of BmHEXIM1, and promoted BmNPV proliferation by repressing NF-κB-driven immune responses. In the mammalian immune system, it was also found that HMBA represses NF-κB-mediated immune pathways [16,49,50].

According to the results of our study, the potential mechanism of the BmHEXIM1-mediated promotion of BmNPV proliferation is summarized in Figure 6. BmHEXIM1 was significantly upregulated after BmNPV infection, while the host antiviral response was activated (Figure 6A). After BmRelish is activated by STING, BmRelish induces the expression of BmSTING and antimicrobial peptides to resist BmNPV infection (Figure 6A). Moreover, the altered subcellular localization of BmHEXIM1 after viral infection may be due to regulation by ubiquitination (Figure 6B). In addition, BmHEXIM1 represses BmRelish-dependent transcription, and thereby inhibits the host BmRelish-dependent antiviral response (Figure 6C). However, in the presence of PMA, the repression effect of BmHEXIM1 was inhibited, and BmRelish-dependent transcription was induced (Figure 6D). Moreover, HMBA may inhibit the transcription of BmRelish-dependent genes by inducing BmHEXIM1 expression (Figure 6E).

## Figures and Tables

**Figure 1 viruses-14-02636-f001:**
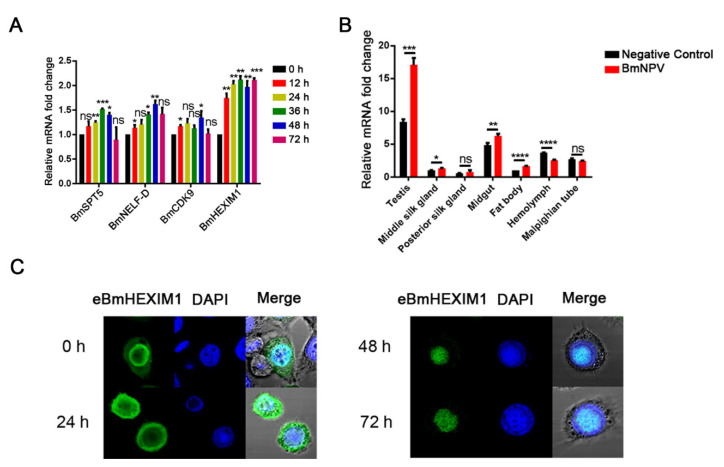
BmHEXIM1 showed a significant response to BmNPV infection. (**A**) The expression levels of BmHEXIM1 in BmN cells at different hours post infection with BmNPV. BmN cells without BmNPV infection as the control. (**B**) The expression levels of BmHEXIM1 in different tissues of silkworm post infection with BmNPV. Fat body of silkworm without BmNPV infection as the control. (**C**) Time course analysis of subcellular localization of BmHEXIM1 by immunofluorescence in BmN cells infected by BmNPV; DAPI, blue; and BmHEXIM1, green. The mRNA level of target genes was normalized to the internal control (BmRPL32). Data represent mean ± SEM of the three independent experiments. The number of asterisks represents the degree of significance with respect to *p*-value. *P*-values were provided as * *p* < 0.05; ** *p* < 0.01; *** *p* < 0.001; **** *p* < 0.0001.

**Figure 2 viruses-14-02636-f002:**
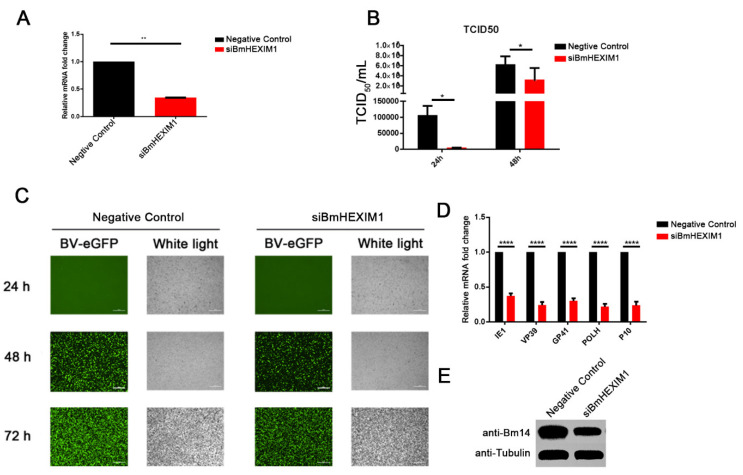
RNAi down-regulated BmHEXIM1 and inhibited the invasion of BmNPV. (**A**) The RNAi of the siBmHEXIM1 RNA decreases the expression level of protein HEXIM1 at 48 h post-transfection. (**B**) TCID_50_ end-point dilution was used to evaluate the production of infectious BV at 24 h and 48 h. (**C**) The infected cells (the cells with the green fluorescence) were observed by an inverted fluorescence microscope at 24 h and 48 h (bar = 100 μm). (**D**) BmN cells were transfected with si-NC and siBmHEXIM1 RNA for 48 h. Then, the cells were infected with BmNPV for 48 h. The viral gene’s expression was measured by qRT-PCR analysis. (**E**) The BmNPV abundance was assessed by analyzing the expression of Bm14 by Western blot after knockdown of BmHEXIM1. The mRNA level of target genes was normalized to the internal control (BmRPL32). Data represent mean ± SEM of the three independent experiments. The number of asterisks represents the degree of significance with respect to *p*-value. *p*-values were provided as * *p* < 0.05; ** *p* < 0.01; **** *p* < 0.0001.

**Figure 3 viruses-14-02636-f003:**
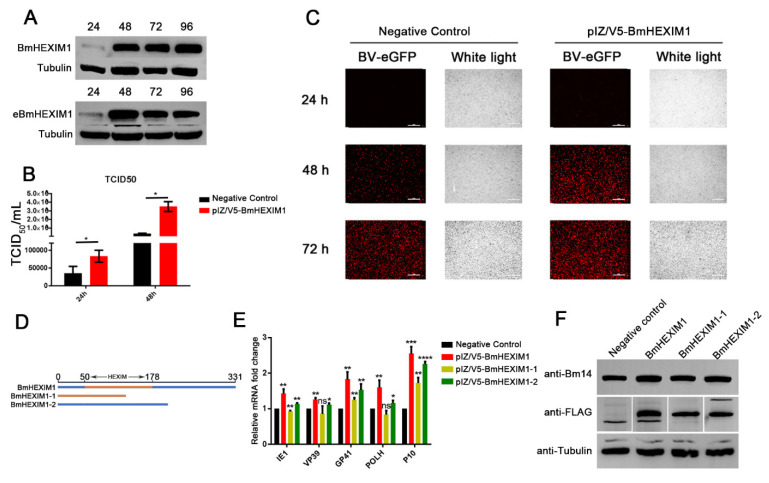
BmHEXIM1 promoted BmNPV proliferation, and its full length was essential to promote viral transcription. (**A**) BmN cells were transfected with pIZ/V5-BmHEXIM1 expressing plasmid, and the expression of BmHEXIM1 was analyzed by Western blot at 24 h, 48 h, 72 h, and 96 h. (**B**) TCID_50_ end-point dilution was used to evaluate the production of infectious BV at 24 h and 48 h. (**C**) The infected cells (the cells with green fluorescence) were observed by an inverted fluorescence microscope at 24 h and 48 h (bar = 100 μm). (**D**) Determination of the region of BmHEXIM1 promoted BmNPV proliferation. (**E**) BmN cells were transfected with pIZ/V5-BmHEXIM1, pIZ/V5-BmHEXIM1-1, and pIZ/V5-BmHEXIM1-2 expressing plasmid for 48 h. Then, the cells were infected with BmNPV for 48 h. The viral gene’s expression was measured by qRT-PCR analysis. (**F**) BmNPV abundance was assessed by analyzing the expression levels of Bm14 by Western blot. The mRNA level of target genes was normalized to the internal control (BmRPL32). Data represent mean ± SEM of the three independent experiments. The number of asterisks represents the degree of significance with respect to *p*-value. *P*-values were provided as * *p* < 0.05; ** *p* < 0.01; *** *p* < 0.001; **** *p* < 0.0001.

**Figure 4 viruses-14-02636-f004:**
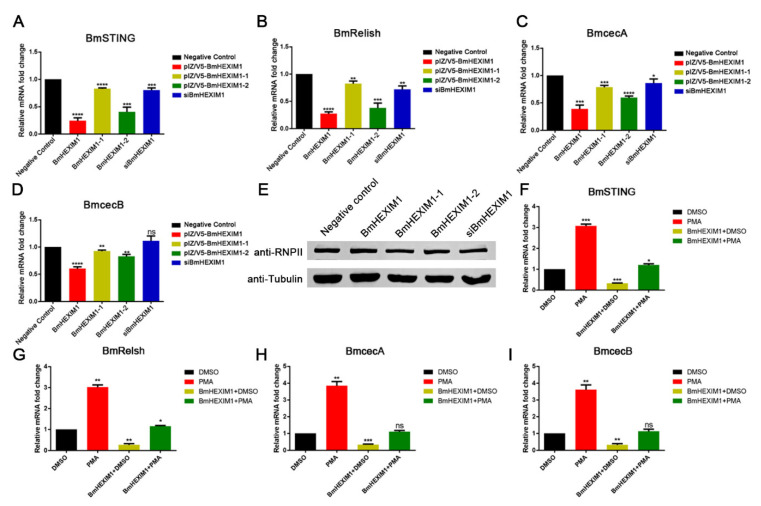
BmHEXIM1 suppresses host innate immunity during BmNPV infection. (**A**–**D**) Quantification of the expression of BmSTING, BmRelish, BmcecA, and BmcecB by qRT-PCR after overexpression of BmHEXIM1, BmHEXIM1-1, BmHEXIM1-2, or knockdown of BmHEXIM1 in BmN cells after BmNPV infection. (**E**) The phosphorylation level of RNP Ⅱ by Western blot after overexpression of BmHEXIM1, BmHEXIM1-1, BmHEXIM1-2, or knockdown of BmHEXIM1 in BmN cells. (**F**–**I**) In the absence or presence of PMA, the expression of BmSTING, BmRelish, BmcecA, and BmcecB was also analyzed by qRT-PCR or by the co-expression of HEXIM1. The mRNA level of target genes was normalized to the internal control (BmRPL32). Data represent mean ± SEM of the three independent experiments. The number of asterisks represents the degree of significance with respect to *p*-value. *p*-values were provided as * *p* < 0.05; ** *p* < 0.01; *** *p* < 0.001; **** *p* < 0.0001.

**Figure 5 viruses-14-02636-f005:**
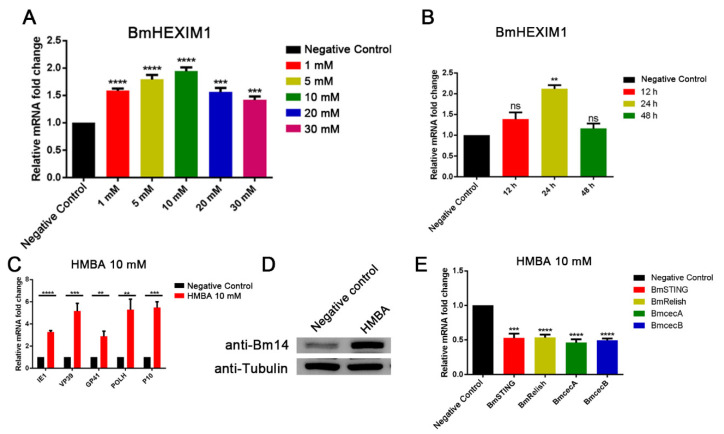
HMBA inhibited BmRelish-mediated pathway and promoted BmNPV proliferation. (**A**) BmN cells were stimulated with increasing concentrations of HMBA (1, 5, 10, 20, and 30 mM) for 24 h, and the expression levels of BmHEXIM1 were measured by qRT-PCR analysis. (**B**) BmN cells stimulated or not with HMBA (10 mM) for 12 h, 24 h, and 48 h and the expression of BmHEXIM1 were measured by qRT-PCR analysis. (**C**) BmN cells stimulated with HMBA (10 mM) for 24 h and the expression of viral genes were measured by qRT-PCR analysis. (**D**) BmN cells stimulated with HMBA (10 mM) for 24 h, and then added BmNPV for 48 h. BmNPV abundance was assessed by analyzing the expression of Bm14 by Western blot. (**E**) BmN cells stimulated with HMBA (10 mM) for 24 h and the expression levels of BmSTING, BmRelish, BmcecA, and BmcecB were measured by qRT-PCR analysis. The mRNA level of target genes was normalized to the internal control (BmRPL32). Data represent mean ± SEM of the three independent experiments. The number of asterisks represents the degree of significance with respect to *p*-value. *p*-values were provided as ** *p* < 0.01; *** *p* < 0.001; **** *p* < 0.0001.

**Figure 6 viruses-14-02636-f006:**
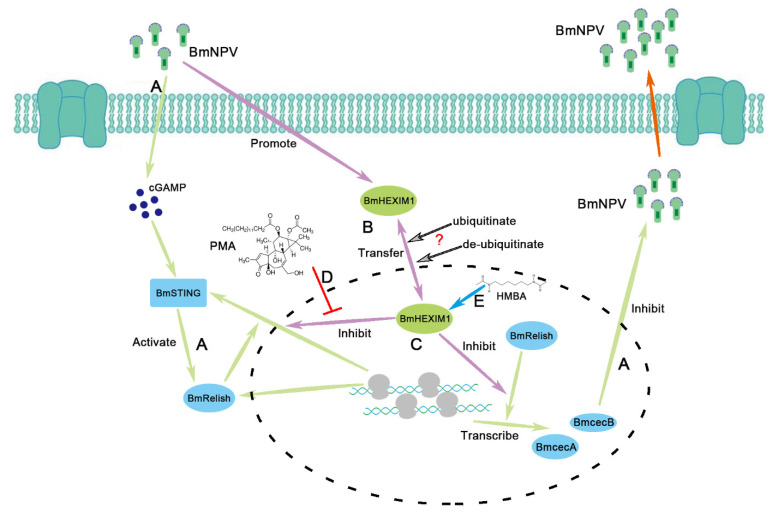
Schematic representation of the putative mechanism(s) of BmHEXIM1 in assisting BmNPV evade antiviral immune response and promoting BmNPV proliferation. (A) BmNPV significantly induced BmHEXIM1 expression, while activating host antiviral pathways, such as STING and Relish. The innate immune pathway mediated by BmRelish can induce the expression of BmSTING antimicrobial peptide genes (BmcecA and BmcecB) to resist BmNPV infection. (B) Altered subcellular localization of BmHEXIM1 may be regulated by ubiquitination following viral infection. “? “ were provided as “not determined”. (C) BmHEXIM1 inhibits BmRelish-dependent transcription, leading to suppression of the host’s antiviral response. (D) Stimulation of PMA inhibits BmHEXIM1 to repress BmRelish-dependent transcription. (E) HMBA treatment induces BmHEXIM1 expression to suppress host antiviral response.

## Data Availability

Not applicable.

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
