# Peer review of "Overexpression *Bombyx mori* HEXIM1 Facilitates Immune Escape of *Bombyx mori* Nucleopolyhedrovirus by Suppressing BmRelish-Driven Immune Responses"

_viruses, 2022, doi:10.3390/v14122636_

Round 1
Reviewer 1 Report
Manuscript ID: viruses- 2028114
Authors: Guanping Chen, Yuedong Li, Xiangshuo Kong, Shudi Zhao, Jiale Li and Xiaofeng Wu
Title: Bombyx mori HEXIM1 facilitates immune escape of Bombyx mori nucleopolyhedrovirus by suppressing BmRelish-driven 3 immune responses
This manuscript describes studies that investigate the insect immune response and anti-viral activities during infection by the prolific baculovirus Bombyx mori nucleopolyhedrovirus (BmNPV), a large DNA virus of lepidopteran insects, specific the silk moth Bombyx mori. The study focuses on the B. mori gene HEXIM1, which in vertebrates is known negatively regulate transcription elongation normally mediated by the positive transcription elongation factor b (P-TEFb). Here the authors identify the B. mori gene encoding HEXIM1 (BmHEXIM1), clone it, and characterize its functions by plasmid transfections of cultured B. mori cells, which are permissive for BmNPV. During infection, it was observed that transcription of BmHEXIM1 is increased (protein levels were not determined). Moreover, by using RNAi knockdown approaches evidence is presented that BmHEXIM1 enhances BmNPV yields. The authors analyze the effect of overexpressed BmHEXIM1 on transcription of B. mori immune genes, including those regulated by Relish, and find that their transcription is reduced, suggesting that BmHEXIM1 represses the anti-viral response to facilitate virus multiplication. These results are interesting and provide insight into host-baculovirus interactions. However, the overall effects are very modest and the reader questions the relevance of these findings, especially in light of the suggestion that BmHEXIM1 functions differently than vertebrate HEXIM1.
Comments:
1. The use of RNAi to knockdown expression of a protein is a useful strategy to test the function of a protein in the context of infection. However, knockdown of BmHEXIM1 reduced virus yields by no more than 2-fold at 48 hours. But, the authors claim that BmHEXIM1 is important in promoting BmNPV multiplication. This reviewer disagrees. It is possible that reducing the protein level of a critical transcription regulator affects cell viability, which would affect virus productivity. Although the authors show that mRNA for BmHEXIM1 is reduced by 2-fold, they have not assessed whether BmHEXIM1 protein levels were affected, which is more important.
2. Overexpression of BmHEXIM1 was achieved by plasmid transfection of cultured B. mori cells. Yields of virus increased modestly in the presence of transfected recombinant BmHEXIM1. It looks like the increase is 3 fold in the graph (Fig. 3) – but the graph is difficult to read and the authors never state the actual difference. Such a small increase is not convincing. What is eBmHEXIM1 (Fig. 3A)? It is not defined in the text.
3. The authors construct internal deletions of BmHEXIM1 to remove potential function domains and then show that these deletion mutations do not enhance virus productivity. But the levels of the mutated proteins are not measured. It is very possible that the deletions are not stable or produced poorly and therefore will not function.
4. The authors conclude that BmHEXIM1 is not involved in regulating viral transcriptional elongation by regulating viral polymerase phosphorylation. They use immunoprecipitations for this purpose. The pulldowns are negative. But there are no positive controls. Thus, it is not clear that the immunoprecipitations functioned here.
5. Fig. 4. The authors test the effect of overexpressed BmHEXIM1 on the transcription of specific B. mori immune genes in cultured cells. In general, there is a reduction of mRNA levels of BmSting, BmRelish, and others. The authors conclude that BmHEXIM1 functions to suppress the immune response during virus infection. However, it is not clear that the same reduction would be observed with the expression of other non-immune genes. Such controls should be included. Surprisingly, RNAi knockdown of endogenously expressed BmHEXIM1 did not affect the same immune response genes. This finding raises doubt about the overexpression results, which can be artifactual.
6. This study uses multiple experimental strategies to assess the effect of BmHEXIM1. However, in no case do the authors report (in the text) the numerical differences. The most common terms used are “significant” or “significantly” to describe the level of difference (there are 18 uses of these terms!). Such subjectivity does little to convince the reader than there is a real difference. State the specific differences in accurate terms, please.
7. Many of the figures are simply too small to be visualized by the reader. Graphs showing quantitation of virus yields lack a Y-axis and the increments are impossible to read. These are major oversights on the part of the authors.
8. The authors should check their manuscript carefully for the use of inaccurate English.
Author Response
Dear Reviewer:
We thank the reviewer for reading our manuscript and we are pleased to improve our manuscript with the help of your thoughtful suggestions. The following is a list of responses to your comments. We sincerely hope that this revised manuscript could address all your comments and suggestions.
Comment 1:
“The use of RNAi to knockdown expression of a protein is a useful strategy to test the function of a protein in the context of infection. However, knockdown of BmHEXIM1 reduced virus yields by no more than 2-fold at 48 hours. But the authors claim that BmHEXIM1 is important in promoting BmNPV multiplication. This reviewer disagrees. It is possible that reducing the protein level of a critical transcription regulator affects cell viability, which would affect virus productivity. Although the authors show that mRNA for BmHEXIM1 is reduced by 2-fold, they have not assessed whether BmHEXIM1 protein levels were affected, which is more important.”
Response 1:
The reviewer raised the main question that is regarding about language and description. Briefly, we have rephrased some description and streamline our presentation of the results and the discussion, which had been extensively revised.
For the RNA interference experiments, we designed 3 target sites, and the siRNA used in the manuscript worked best. However, our lab did not have BmHEXIM1 antibody, and antibodies to homologous genes of other species are less specific in the BmN cells, so we could not assess the effect of siRNA knockdown of endogenous BmHEXIM1 at the protein level. In addition, we also performed RNA-seq and showed that knockdown of BmHEXIM1 had no effect on most cellular gene expression, but had an effect on some genes involved in amino acid metabolism during viral infection (data unpublished), such as KMO and BCAT.
Comment 2:
“Overexpression of BmHEXIM1 was achieved by plasmid transfection of cultured B. mori cells. Yields of virus increased modestly in the presence of transfected recombinant BmHEXIM1. It looks like the increase is 3-fold in the graph (Fig. 3) – but the graph is difficult to read and the authors never state the actual difference. Such a small increase is not convincing. What is eBmHEXIM1 (Fig. 3A)? It is not defined in the text.”
Response 2:
Thanks for the comment of reviewers. Your suggestion is very important to our manuscript. We have already corrected all the figures and stated the actual differences in the [revised manuscript]. Overexpression of BmHEXIM1 resulted in a 2.3-fold increase in BV titers at 24h and a 9.8-fold increase at 48h.
Comment 3:
“The authors construct internal deletions of BmHEXIM1 to remove potential function domains and then show that these deletion mutations do not enhance virus productivity. But the levels of the mutated proteins are not measured. It is very possible that the deletions are not stable or produced poorly and therefore will not function.”
Response 3:
The authors would like to thank the reviewer for the suggestion. We're already on Fig3. F added result of the levels of the mutated proteins which were measured by Western Blot. All experiments were performed under conditions that ensure stable expression of all mutants within BmN cells.
Comment 4:
“The authors conclude that BmHEXIM1 is not involved in regulating viral transcriptional elongation by regulating viral polymerase phosphorylation. They use immunoprecipitations for this purpose. The pulldowns are negative. But there are no positive controls. Thus, it is not clear that the immunoprecipitations functioned here.”
Response 4:
We totally understand the reviewer’s concern. We have modified the [revised manuscript] by adding the results of positive control experiments in Fig. S1B, and the results indicate that BmHEXIM1 did not interact with viral RNA polymerase and PK1. Therefore, we concluded that BmHEXIM1 is not involved in the regulation of viral polymerase phosphorylation.
Comment 5:
“Fig. 4. The authors test the effect of overexpressed BmHEXIM1 on the transcription of specific B. mori immune genes in cultured cells. In general, there is a reduction of mRNA levels of BmSting, BmRelish, and others. The authors conclude that BmHEXIM1 functions to suppress the immune response during virus infection. However, it is not clear that the same reduction would be observed with the expression of other non-immune genes. Such controls should be included. Surprisingly, RNAi knockdown of endogenously expressed BmHEXIM1 did not affect the same immune response genes. This finding raises doubt about the overexpression results, which can be artifactual.”
Response 5:
The authors would like to thank the reviewer for the suggestion. As shown in the [revised manuscript] Fig. 4E, overexpression of both BmHEXIM1 and the mutant protein, did not affect normal host transcription by inhibiting RNA polymerase phosphorylation. Furthermore, we fully understand your concern that RNAi knockdown endogenous expression of BmHEXIM1 has no effect on the immune response. We were also surprised by this result. However, in mammalian cell line, NF-κB is dose-dependently inhibited by exogenous HEXIM1, and NF-κB was not affected by normal endogenous HEXIM1 expression[1]. Therefore, the result that RNAi knockdown of endogenous expression of BmHEXIM1 did not affect the immune response was acceptable. To better present our results, we have modified the title in the [revised manuscript].
[1] Ouchida R, Kusuhara M, Shimizu N, Hisada T, Makino Y, Morimoto C, Handa H, Ohsuzu F, Tanaka H. Suppression of NF-kappaB-dependent gene expression by a hexamethylene bisacetamide-inducible protein HEXIM1 in human vascular smooth muscle cells. Genes Cells. 2003 Feb;8(2):95-107. doi: 10.1046/j.1365-2443.2003.00618.x. PMID: 12581153.
Comment 6:
“This study uses multiple experimental strategies to assess the effect of BmHEXIM1. However, in no case do the authors report (in the text) the numerical differences. The most common terms used are “significant” or “significantly” to describe the level of difference (there are 18 uses of these terms!). Such subjectivity does little to convince the reader than there is a real difference. State the specific differences in accurate terms, please.”
Response 6:
Thank you so much for your careful check. Briefly, we have rephrased some description and refined our presentation of the results in the [revised manuscript], which had been extensively revised.
Comment 7:
“Many of the figures are simply too small to be visualized by the reader. Graphs showing quantitation of virus yields lack a Y-axis and the increments are impossible to read. These are major oversights on the part of the authors.”
Response 7:
We are very sorry for our careless mistake and all figures were rectified. In addition, we provide higher quality original figures in the [revised manuscript].
Comment 8:
“The authors should check their manuscript carefully for the use of inaccurate English.”
Response 8:
Thank you for your valuable and thoughtful comments. We have carefully checked and improved the English writing in the [revised manuscript].
Reviewer 2 Report
In this manuscript, the authors cloned and characterized BmHEXIM1, and examined the role of BmHEXIM1 in BmNPV life cycle. The authors demonstrated that BmHEXIM1 increased the proliferation of BmNPV, and its full length is essential for assisting BmNPV immune escape by suppressing BmRelish-driven immune responses. It is very interesting study, but the manuscript has several weaknesses that need to be addressed in its current form to make it suitable for publication in the journal. Listed below are my specific comments.
1. In Fig. 4, knocking down BmHEXIM1 did not affect BmRelish expression, but inhibited viral proliferation in Fig. 2. However, the authors did not explain this phenomenon in the discussion section, and it needs to be added.
2. In Fig. 5B, the Negative control marker is missing, and the "negative control" of all figures is misspelled.
3. The manuscript contains some grammatical errors and needs to be further refined.
(1) Line 59, “HEXIM1 was shown to bind the NEAT1 non-coding nuclear RNA to forming a new RNA protein complex” should be modified to “HEXIM1 was shown to bind the NEAT1 non-coding nuclear RNA to form a new RNA-59 protein complex.”
(2) Line 66, “a series of experiment were done” should be modified to “a series of experiments were done.”
(3) Line 211, “Knockdown of BmHEXIM1” should be modified to “knockdown of BmHEXIM1.”
(4) Line 287, “and also regulate NF-κB” should be modified to “and also regulates NF-κB.”
(5) Line 395, “and thereby inhibited the host” should be modified to “and thereby inhibits the host.”
Author Response
Dear Reviewer:
We thank the reviewer for reading our manuscript and we are pleased to improve our manuscript with the help of your thoughtful suggestions. The following is a list of responses to your comments. We sincerely hope that this revised manuscript could address all your comments and suggestions.
Comment 1:
“In Fig. 4, knocking down BmHEXIM1 did not affect BmRelish expression, but inhibited viral proliferation in Fig. 2. However, the authors did not explain this phenomenon in the discussion section, and it needs to be added.”
Response 1:
Thanks for your great suggestion on improving the accessibility of our manuscript. In accordance with this comment, we have added discussion in the Discussion section.
Comment 2:
“In Fig. 5B, the Negative control marker is missing, and the "negative control" of all figures is misspelled.”
Response 2:
We are very sorry for our careless mistake and all figures were rectified in the [revised manuscript].
Comment 3:
“The manuscript contains some grammatical errors and needs to be further refined.”
(1) Line 59, “HEXIM1 was shown to bind the NEAT1 non-coding nuclear RNA to forming a new RNA protein complex” should be modified to “HEXIM1 was shown to bind the NEAT1 non-coding nuclear RNA to form a new RNA-59 protein complex.”
(2) Line 66, “a series of experiment were done” should be modified to “a series of experiments were done.”
(3) Line 211, “Knockdown of BmHEXIM1” should be modified to “knockdown of BmHEXIM1.”
(4) Line 287, “and also regulate NF-κB” should be modified to “and also regulates NF-κB.”
(5) Line 395, “and thereby inhibited the host” should be modified to “and thereby inhibits the host.”
Response 3:
Thank you for the good suggestion. We have carefully and thoroughly proofread the manuscript to correct all the grammar and typos.
Reviewer 3 Report
Well-designed and planned study; however, almost all the figures must be improved in resolution and readability. Statistics on histograms are confusing and should be indicated according to which comparison the statistical significances are presented.
Alignment has to be improved, and relevant domains have to be shown. The resolution of the microscopy images is not acceptable. 10X and 40X images have to be provided with 300dpi resolution (minimum). Overall all the figures have to be improved for optimal quality.
Author Response
Dear Reviewer:
We thank the reviewer for reading our manuscript and we are pleased to improve our manuscript with the help of your thoughtful suggestions. The following is a list of responses to your comments. We sincerely hope that this revised manuscript could address all your comments and suggestions.
Comment 1:
“Well-designed and planned study; however, almost all the figures must be improved in resolution and readability. Statistics on histograms are confusing and should be indicated according to which comparison the statistical significances are presented.
Alignment has to be improved, and relevant domains have to be shown. The resolution of the microscopy images is not acceptable. 10X and 40X images have to be provided with 300dpi resolution (minimum). Overall all the figures have to be improved for optimal quality.”
Response 1:
We are very sorry for our careless mistake and all figures were rectified. In addition, we provided higher quality original figures in the [revised manuscript].
Round 2
Reviewer 3 Report
Figure 1 has not been improved after revision, and the resolution is insufficient to visualize the date. Other statistics also suboptimally improved, especially microscopy images.
Author Response
Dear Reviewer:
We thank the reviewer for reading our revised manuscript and we are pleased to improve our revised manuscript with the help of your thoughtful suggestions. The following is a list of responses to your comments. We sincerely hope that this revised manuscript could address all your comments and suggestions.
Comment 1:
“Figure 1 has not been improved after revision, and the resolution is insufficient to visualize the date.”
Response 1:
We are very sorry for our careless mistake and Figure 1 were rectified. For a better view of the data, we have reformatted Figure 1 in the [revised manuscript]. In this case, we show C, D, and E of the original Figure 1 in the supplemental Figures to visualize the date more clearly.
Comment 2:
“Other statistics also suboptimally improved, especially microscopy images.”
Response 2:
Thank you for your valuable and thoughtful comments. We have carefully checked and improved statistics in the [revised manuscript].